# Impact of Reduced Saliva Production on Intestinal Integrity and Microbiome Alterations: A Sialoadenectomy Mouse Model Study

**DOI:** 10.3390/ijms252212455

**Published:** 2024-11-20

**Authors:** Kanna Maita, Hisako Fujihara, Mitsuki Matsumura, Moeko Miyakawa, Ryoko Baba, Hiroyuki Morimoto, Ryoko Nakayama, Yumi Ito, Koji Kawaguchi, Yoshiki Hamada

**Affiliations:** 1Department of Oral and Maxillofacial Surgery, School of Dental Medicine, Tsurumi University, 2-1-3 Tsurumi, Tsurumi-ku, Yokohama 230-8501, Japan; pd21009@stu.tsurumi-u.ac.jp (K.M.); hamada-y@tsurumi-u.ac.jp (Y.H.); 2Department of Oral Hygiene, Tsurumi Junior College, 2-1-3 Tsurumi, Tsurumi-ku, Yokohama 230-8501, Japan; 3Department of Anatomy, School of Medicine, University of Occupational and Environmental Health, 1-1, Iseigaoka, Yahatanishi, Kitakyushu 807-8555, Japan; 4Department of Pathology, School of Dental Medicine, Tsurumi University 2-1-3 Tsurumi, Tsurumi-ku, Yokohama 230-8501, Japan; 5Department of Diagnostic Pathology, Tsurumi University Dental Hospital, Yokohama 230-8501, Japan

**Keywords:** sialoadenectomy, xerostomia, gastrointestinal tract, microbiota, EGF, PARP

## Abstract

This study investigates the effect of reduced saliva production on intestinal histological structure and microbiome composition using a sialoadenectomy murine model, evaluating differences in saliva secretion, body weight, intestinal histopathological changes, and microbiome alteration using 16S rRNA gene sequencing across three groups (control, sham, and sialoadenectomy). For statistical analysis, one-way analysis of variance and multiple comparisons using Bonferroni correction were performed. *p*-values < 0.05 were considered statistically significant. Microbiome analysis was performed using Qiime software. The results show that reduced saliva secretion leads to structural changes in the intestinal tract, including shorter and atrophic villi, deformed Paneth cells, decreased goblet cell density, and immunohistochemical changes in epidermal growth factor and poly(ADP-ribose) polymerase-1, especially at three months after surgery. They also showed significant alterations in the intestinal microbiome, including increased *Lactobacillaceae* and altered populations of *Ruminococcaceae* and *Peptostreptococcaceae*, suggesting potential inflammatory responses and decreased short-chain fatty acid production. However, by 12 months after surgery, these effects appeared to be normalized, indicating potential compensatory mechanisms. Interestingly, sham-operated mice displayed favorable profiles, possibly due to immune activation from minor surgical intervention. This study underscores saliva’s essential role in intestinal condition, emphasizing the “oral–gut axis” and highlighting broader implications for the relationship between oral and systemic health.

## 1. Introduction

Recent research underscores the essential roles of oral and intestinal microbiomes in maintaining human health, influencing both local and systemic physiology. Emerging evidence has highlighted the “oral–gut axis”, suggesting that the oral microbiome not only reflects oral health but may influence intestinal health and related diseases. This is particularly relevant in inflammatory diseases like inflammatory bowel disease (IBD), where dysbiosis in both the oral and gut microbiomes contribute to disease pathogenesis [1,2]. For example, periodontal pathogens such as *Porphyromonas gingivalis* can alter intestinal microbiota, enhancing systemic inflammation and metabolic issues [3,4]. Such findings indicate that oral microbiota may play a critical role in relation to intestinal dysbiosis, impacting immune responses and compromising the gut barrier.

Clinically, patients with compromised salivary function, such as those with head and neck cancer undergoing radiation therapy [5,6] or those with Sjögren syndrome [7], frequently present with IBD-like symptoms, suggesting multifactorial pathological involvement. However, the precise relationship between reduced saliva production, salivary gland dysfunction, and gastrointestinal symptoms remains insufficiently understood. Numerous conditions, including Sjögren syndrome [8], postoperative symptoms following sialoadenectomy for salivary tumors [9], radiation therapy for head and neck cancer [10], aging [11], and diabetes mellitus [12], disrupt salivary flow and induce glandular dysfunction. These conditions frequently result in xerostomia (equivalent to dry mouth) and are associated not only with swallowing difficulties [11] but also with gastrointestinal symptoms such as nausea, epigastric pain, constipation, and diarrhea [13]. Salivary dysfunction compromises the transfer of oral microbiota to the gastrointestinal tract, potentially destabilizing gut homeostasis and promoting inflammatory and metabolic disorders [2]. Nutrient absorption mainly occurs in the small intestine, where it is facilitated by a polarized epithelial layer [14]. Gastrointestinal dysfunction can impair nutrient absorption, leading to mucosal atrophy and gut microbiome alteration [15,16]. The intestinal microbiome, primarily considered to be involved in nutrient absorption in the small intestine, has gained attention as a significant factor in both gastrointestinal and systemic diseases.

For instance, dysbiosis has been observed in patients with irritable bowel syndrome (IBS), who exhibit elevated levels of *Proteobacteria*, *Lactobacillaceae*, and *Bacteroides* alongside decreased *Ruminococcaceae* and *Tannerellaceae* compared to healthy individuals [17]. Additionally, specific intestinal microbiota, including *Bacteroides fragilis*, *Streptococcus gallolyticus*, *Enterococcus faecalis*, and *Escherichia coli*, are implicated in colorectal cancer [18].

Beyond gastrointestinal disorders, intestinal dysbiosis has been reported to link to various systemic conditions such as allergic diseases [19], central nervous system diseases [20], abdominal aortic aneurysm [21], obesity [22], and diabetes mellitus [12,23], establishing the microbiome as a functional organ integral to multiple physiological systems.

This study aims to explore the impact of reduced saliva on intestinal conditions using sialoadenectomy-induced xerostomia model in mice. To our best knowledge, this is the first direct investigation of how saliva reduction affects intestinal integrity and microbiota composition by examining alteration in intestinal microbiota and histopathological changes.

## 2. Results

### 2.1. Mice in the Sialoadenectomy Group Showed Regular Body Weight Gain and Significantly Lower Saliva Secretion

The average body weight of mice in all groups before surgical treatment was 21.1 ± 0.93 g. The sialoadenectomy group exhibited a slower increase in body weight compared with the other two groups 6 months after surgery. However, this difference did not reach statistical significance during the 12-month experimental period (Figure 1a). Before surgical treatment, the average saliva secretion in 30 min for the mice in all groups was 0.26 ± 0.03 μL. After surgical treatment and throughout the 12-month experimental period, there was a significant decrease in saliva secretion in the sialoadenectomy group compared with the other two groups (*p* < 0.01) (Figure 1b).

### 2.2. Effects of Decreased Saliva Secretion Due to Sialoadenectomy on Histopathological Changes in the Intestinal Tract with Hematoxyline and Eosin Staining

#### 2.2.1. Morphological Changes in Intestinal Villi

We compared the histopathological changes in the intestinal tract of the three groups at 3, 6, and 12 months after surgery (Figure 2).

To assess morphological changes in villi, we conducted a comprehensive analysis of villus length, thickness, and cross-sectional area in the jejunum and ileum at 3, 6, and 12 months after surgery. Analysis of villus length revealed that the sialoadenectomy group exhibited significantly shorter villi in the jejunum at 3 months and in the ileum at 3 and 6 months (*p* < 0.05). In contrast, the sham group showed a significant increase in ileal villus length at 12 months after surgery (Table 1).

Villus thickness presented variable results across the three groups. The control group displayed significantly thicker jejunal villi at 3 months (*p* < 0.01), the sham group showed increased ileal villus thickness at 6 months (*p* < 0.05), and the sialoadenectomy group demonstrated thicker jejunal villi at 12 months after surgery (*p* < 0.01) (Table 2). No significant difference in the thickness of villi was observed at other time points.

On the contrary to the inconsistent results observed in villus thickness, the sialoadenectomy group exhibited significantly smaller area of sectioned jejunal villi compared to the other two groups at 3 months after surgery (*p* < 0.01) (Table 3), offering more precise representation of morphological changes in villus structure than isolated measurements of length or thickness. No significant difference was observed in this group at other time points.

#### 2.2.2. Morphological Changes in Paneth Cells

High-power microscopic examination revealed deformed Paneth cells in the sialoadenectomy group at 3 months after surgery, which were restored at 6 and 12 months. The sham group also demonstrated a similar pattern to that of the sialoadenectomy group. Unexpectedly, the control group showed the deformation of Paneth cells in the jejunum and ileum at 12 months (Figure 3).

### 2.3. Sialoadenectomy Group Demonstrated Significantly Reduced PAS-Positive Goblet Cell Density, Decreased Expression of EGF, and Increased Expression of PARP-1 at 3 Months After Surgery

#### 2.3.1. The Result of d-PAS-Positive Goblet Cell Number and Cell Density per Unit Area

We performed d-PAS staining to evaluate the presence of goblet cells within the jejunal and ileal villi (Figure 4a). At 3 months after surgery, the number of d-PAS-positive goblet cells was significantly reduced in the sialoadenectomy group compared with the other two groups. However, there were no significant differences in the jejunum at 6 and 12 months after surgery or in the ileum at 12 months after surgery (Table 4).

Using the previously calculated villi area (Table 3), we quantified goblet cell density per unit area (10,000 μm^2^) and found a significant reduction in goblet cell density in the ileum at 3 months after surgery. No significant differences were observed in the jejunum across all time points nor in the ileum at 6 and 12 months after surgery (Figure 4b).

#### 2.3.2. Immunohistochemical Analysis of EGF, VEGF, and PARP-1

We also performed immunohistochemical analysis for epidermal growth factor (EGF), Vascular Endothelial Growth Factor (VEGF), and poly(ADP-ribose) polymerase-1(PARP-1) to investigate morphological changes in the villi. At 3 months after surgery, EGF expression was strongly positive on the surface mucosa in the jejunum, ileum, and colon in the control group compared with the other two groups (Figure 5a). However, this difference was not statistically significant at 6 and 12 months after surgery (Appendix A). The quantification of DAB staining by conversion to grayscale revealed that the grayscale value in the control group was significantly smaller than in the other two groups (Figure 5b), meaning stronger positivity in the control group compared with the two other groups. VEGF expression (Appendix A) remained strongly positive across all groups throughout the 12-month experimental period, with no significant differences observed in the semi-quantitative analysis. At 3 months after surgery, PARP expression was significantly positive on the surface of the colon and inside the jejunal and ileal villi in the sialoadenectomy group (Figure 6). Similarly to the EGF expression pattern, this difference was not statistically significant at 6 and 12 months after surgery (Appendix A).

### 2.4. Intestinal Microbiome Analysis

#### 2.4.1. Microbiome Composition Was Significantly Different 3 Months After Surgery in the Sialoadenectomy Group and Similar Among All Groups at 6 and 12 Months After Surgery

The number of intestinal microbiomes with relative abundances > 5% and >0.1% in the three groups did not show significant difference (Appendix A).

At the phylum level, Firmicutes were the most prevalent, followed by Bacteroidota. Together, these two phyla consistently comprised >85% of the microbiome in all three groups without any significant differences (Figure 7a). At the genus level, several bacteria showed significant differences in the mean relative abundance as follows:f_*Lactobacillaceae*; g_*Lactobacillus*: significantly higher in the sialoadenectomy group (mean relative abundance: 11.99%) at 3 months after surgery and significantly higher in the sham group (mean relative abundance: 12.50%) at 6 months after surgery.f_*Peptostreptococcaceae*; g_*Romboutsia*: significantly higher in the control group (mean relative abundance: 4.96%) at 3 months after surgery and significantly lower in the sham group (mean relative abundance: 0.0%) at 6 months after surgery.f_*Tannerellaceae*; g_*Parabacteroides*: significantly lower in the control group than in the sham and sialoadenectomy groups at 3 months after surgery (mean relative abundance: 0.54%, 1.36%, and 1.07%, respectively). At 6 months after surgery, the control and sialoadenectomy groups were significantly lower than the sham group (mean relative abundance: 0.43%, 0.54%, and 1.29%, respectively).f_*Ruminococcaceae*; g_*incertae*_*sedis*: significantly higher in the control group than the sham and sialoadenectomy groups at 3 months after surgery (mean relative abundance: 1.80%, 016%, and 0.11%, respectively) (Figure 7b).

Within f_*Ruminococcaceae*, which was significantly abundant in the control group, there were several genera, including f_*Ruminococcaceae*; g_*Eubacterium siraeum,* and f_*Ruminococcaceae*; g_uncultured, with significantly higher abundance in the control group and decreased abundance in the sialoadenectomy group 3 months after surgery. Additionally, >98% of genera belonging to f_*Ruminococcaceae* were substantially similar in the microbiome 12 months after surgery across all three groups, except f_*Ruminococcaceae*; g_Ruminococcus, which comprised approximately 2% of f_*Ruminococcaceae* and demonstrated a significant increase in abundance in the sham and sialoadenectomy groups at 12 months after surgery.

#### 2.4.2. Within-Subject α-Diversity Showed Similar Trends and Between-Subject β-Diversity Showed Different Patterns of Significant Differences at 3, 6, and 12 Months After Surgery Within the Three Groups

The three groups did not exhibit significant differences in α-diversity at 3, 6, and 12 months after surgery. Although the average observed features for α-diversity were approximately doubled at 12 months compared with 3 and 6 months after surgery, significant differences were not observed across all time points (Figure 8).

Figure 9 shows the β-diversity of microbial communities in each group at 3, 6, and 12 months after surgery. The sialoadenectomy group showed significant interindividual differences compared with the control group at 3 months after surgery (*p* < 0.05). At 6 months after surgery, the sham group showed significant interindividual differences compared with the control and sialoadenectomy groups (*p* < 0.05). No significant differences were observed among the three groups at 12 months after surgery.

#### 2.4.3. LEfSe Analysis Identified Specific Microbiome Alterations After Sialoadenectomy

We applied the LEfSe algorithm to identify characteristic bacterial taxa where the abundance was significantly affected by sialoadenectomy. Three months after surgery, the sialoadenectomy group exhibited a unique microbiome highly abundant in o_*Lactobacillales*, f_*Lactobacillaceae*, and g_*Lactobacillus* at each taxonomic hierarchy, respectively. The sham group showed a high LDA score for g_*Alistipes* (within o_*Bacteroides*; f_*Rikenellaceae*). In contrast, the control group showed high LDA scores for o_*Oscillospirales*; f_*Ruminococcaceae*; g_*Romboutsia*, o_*Peptostreptococcales-Tissierellales*; f_*Peptostreptococcaceae*, and o_*Erysipelotrichales*; f_*Erysipelotrichaceae*; g_*Turicibacter* (Figure 10a, Figure 11a and Appendix A).

Six months after surgery, the sialoadenectomy group was characterized by a significant dominance of c_Bacilli. The sham group showed high LDA scores for p_*Proteobacteria*; c_*Gammaproteobacteria*; o_*Enterobacterales*; f_*Enterobacteriaceae*, p_*Deferribacterota* (Deferribacteres); c_*Deferribacteres*; o_*Deferribacterales*; f_*Deferribacteraceae*; g_*Mucispirillum*; s_*Mucispirillum schaedleri*, and f_*Oscillospiraceae*. The control group showed high LDA scores for p_*Firmicutes*, g_*Romboutsia*, o_*Peptostreptococcales-Tissierellales*; f_*Peptostreptococcaceae*, and o_*Erysipelotrichales*; f_*Erysipelotrichaceae*; g_*Turicibacter* (Figure 10b, Figure 11b and Appendix A).

Twelve months after surgery, the number of microbiomes with high LDA scores had decreased. No significant microbiome was identified in the Sialoadenectomy group, while each sham and control group exhibited two microbiomes: f_*S24-7* and f_*Clostridiaceae* in sham group, and o_Turicibacterales; f_*Turicibacteraceae* (synonymous with *Erysipelotrichaceae*) and g_*Turicibacter* in control group (Figure 10c, Figure 11c and Appendix A).

#### 2.4.4. Heatmap Analysis Showed a Microbiome Substantially Similar to That of the LEfSe Analysis

We performed a heatmap analysis to visualize the relative abundance of microbiomes across the three groups. The results were consistent with those obtained from the LEfSe analysis.

Three months after surgery, the sialoadenectomy group showed a high relative abundance of f_Lactobacillaceae; g_*Lactobacillus*, f_Aerococcaceae; g_*Aerococcus*, and f_Oscillospiraceae; g_*NK4A214*_group. The control group was significantly abundant in f_Enterobacteriaceae; g_*Enterobacter*, f_Peptostreptococcaceae; g_*Romboutsia*, f_Lachnospiraceae; g_ [*Eubacterium*]_*ventriosum*_group, f_Erysipelotrichaceae; g_*Turicibacter*, f_Ruminococcaceae; g_ [Eubacterium]_*siraeum*_group, and f_Ruminococcaceae; g_*incertae*_*sedis*. Six months after surgery, f_Erysipelotrichaceae; g_*Turicibacter* and f_Peptostreptococcaceae; g_*Romboutsia* were as significantly abundant as at 3 months after surgery, while f_Leuconostocaceae; g_*Weissella* was significantly abundant in the sialoadenectomy group (Appendix A). The sham group did not exhibit any particular taxonomy. Table 5 summarizes the commonly upregulated microbiomes identified in composition ratio, LEfSe, and heatmap analysis by group and time point.

## 3. Discussion

In this study, we investigated the effects of sialoadenetomy on the intestinal tract through comprehensive histopathological and microbiome analysis using murine models.

Histopathologically, the sialoadenectomy group exhibited a significantly changed intestinal structure, such as shortened and atrophic villi (Figure 2, Table 1, Table 2 and Table 3), deformed Paneth cells (Figure 3), decreased density of PAS-positive goblet cells (Figure 4), and altered EGF and PARP-1 immunohistochemical profiles, especially prominent at 3 months after surgery. These changes could have resulted from reduced saliva and its components, which are inevitable for maintaining mucosal integrity. Saliva contains various antibacterial agents including lysozymes, lactoferrin, IgA, and mucins, which disrupt bacterial cell walls, inhibit biofilm formation, prevent bacterial adhesion, and block attachment to mucosal surfaces, respectively [24,25]. Additionally, saliva contains growth factors such as EGF and VEGF, essential for epithelial maintenance [26].

In our study, EGF expression was significantly higher in the control group at 3 months after surgery, followed by insignificance after 6 and 12 months (Figure 5). VEGF expression remained consistently positive across all groups, with no significant differences observed throughout the entire experimental period. Considering that both EGF and VEGF are present in saliva [26], we had predicted that their expression levels would decrease in the sialoadenectomy group. However, only EGF expression was decreased in the sialoadenectomy group at 3 months after surgery (Figure 5), while VEGF levels did not exhibit any significant changes (Appendix A).

The expression level of EGF is significantly higher in healthy individuals [27], and EGF has a crucial role in the intestinal tract as an epithelial mucosa regulator, supporting both intestinal barrier integrity and permeability [28], cell proliferation, and the inhibition of cellular apoptosis [29]. Reduced saliva and salivary EGF levels could lower serum EGF levels [30], potentially affecting EGF level in villi and contributing to atrophic changes in the sialoadenectomy group.

VEGF is primarily known for regulating angiogenesis and promoting the proliferation of intestinal stem and progenitor cells. Decreased VEGF level is associated with conditions such as necrotizing enterocolitis [31]. The direct relationship between decreased saliva and VEGF levels in serum or intestinal tissue remains unclear. Considering VEGF’s rapid response to tissue injury [31], it is possible that intestinal VEGF levels initially declined immediately after surgery and subsequently recovered by 3 months after surgery, resulting in undetectable significant changes within our observed timepoints.

The sialoadenectomy group also exhibited a reduced goblet cell density in the ileum at 3 months post-surgery (Figure 4). Goblet cells are crucial for the maintenance and differentiation of the intestinal mucosal cells [32], and the number of goblet cells is decreased in IBD patients [33]. Considering that goblet cell density is affected by EGF [34] and VEGF [35], our findings of decreased goblet cell density, alongside significantly decreased EGF, are consistent with previous evidence that EGF supports goblet cell density [34], although VEGF contribution was undetermined.

Villi structure changes and increased PARP-1 positivity at 3 months in the sialoadenectomy group (Figure 2 and Figure 6) suggest that apoptotic cell death may occur in the intestines. PARP-1, the most abundant enzyme in the PARP family, catalyzes poly(ADP-ribosyl)ation using nicotinamide adenine dinucleotide as a substrate, thereby influencing various biological activities. Since the role of PARP-1 in DNA repair and cell death following ROS-induced injury [36], ischemia–reperfusion injury [36], and inflammatory injury [37] has been reported and established as a common concept, it is plausible that sialoadenectomy-induced damage to intestinal villi at 3 months after surgery may be attributed to PARP-1 activation. Paneth cells, essential for antimicrobial peptide production, were significantly deformed in the sialoadenectomy group (Figure 3), potentially increasing vulnerability to dysbiosis and inflammation [38].

The microbiome analysis revealed distinct dominant microbiota for each group at each time point (Table 5). Each dominant microbiome has distinct functions. For instance, *Lactobacillaceae*, significantly higher in the sialoadenectomy group at 3 months after surgery, is generally considered beneficial for intestinal health, contributing to disease prevention, short-chain fatty acid (SCFA) production, vitamin metabolism, iron absorption, and immune modulation [39], but excessive abundance has an association with IBS [40]. Conversely, microbiome abundance in the control group at 3 months after surgery, including of o_Oscillospirales; f_*Ruminococcaceae*; g_*Romboutsia*, o_*Peptostreptococcales-Tissierellales*; f_*Peptostreptococcaceae*, and f_*Ruminococcaceae*; g_*incertae_sedis*, is critical for maintaining a healthy intestinal environment, modulating the immune system [41,42,43], and producing SCFAs [44]. Dysbiosis of these microbiotas has been linked to systemic diseases [45,46]; therefore, the histopathological changes and microbiota alterations observed in this study appear to be consistent.

Over the 12-month experiment, the differences in histopathological and microbiota characteristics among the three groups became insignificant, with the recovery of EGF expression and structure of goblet and Paneth cell, and increased α-diversity. This suggests that the impact of sialoadenectomy was strongest at 3 months after surgery, significantly influencing physical conditions. Although the exact pathway remains unclear, we hypothesize that a homeostatic response may have mitigated the effects of sialoadenectomy because Paneth cells, which play a homeostatic role in maintaining intestinal microbiota [38,47,48], showed recovery at 6 and 12 months after surgery (Figure 3).

Unexpectedly, the sham group exhibited more favorable histopathological and microbiome results compared with the control group, particularly at the later experimental time points. LEfSe analysis at 6 months after surgery showed an increase in p_*Proteobacteria*; c_*Gammaproteobacteria*; o_*Enterobacterales*; f_*Enterobacteriaceae* and p_*Deferribacterota* (Deferribacteres); c_*Deferribacteres*; o_*Deferribacterales*; f_*Deferribacteraceae*; g_*Mucispirillum*; s_*Mucispirillum schaedleri* and a decrease in f_*Peptostreptococcus* (Figure 10b and Figure 11b). The two increased microbiomes at the family level contribute to vitamin production, amino acid metabolism [49], iron metabolism [50], and iron transport system [46,51]. *Peptostreptococcus*, which was reduced at 6 months after surgery, is associated with various infectious diseases [52]. Thus, mice in the sham group appeared to be well-nourished with adequate mineral intake and potentially avoided inflammatory conditions, leading to the sham group’s favorable profile. Moreover, minor surgical intervention in the sham group may have activated immune pathways [53], indirectly benefiting intestinal conditions compared to control group.

We selected the sialoadenectomy model over other options, such as Sjögren syndrome [54] or small animal radiation platforms [55], to avoid confounding factors and to directly evaluate the effects of saliva decrease on intestinal condition. Sjögren syndrome models may involve autoimmune responses, including celiac disease or gluten hypersensitivity, complicating interpretation of gastrointestinal symptoms [54]. Radiation models carry risks of collateral tissue damage and chronic inflammation, potentially confounding gastrointestinal assessments [56]. The used sialoadenectomy murine model allowed us to directly examine saliva’s role without additional inflammatory or autoimmune confounders.

We initially hypothesized that the food intake in the sialoadenectomy group would decrease due to decreased saliva production, leading to poorer nutrition and reduced weight gain. Contrary to predictions, this group exhibited regular weight gain with no significant differences compared to the other groups (Figure 1). This outcome may be explained by the paste diet use, facilitating feeding despite the significantly reduced saliva secretion, as previously reported [56].

A limitation of this study is that we were unable to evaluate saliva components or oral microbiome changes due to insufficient saliva secretion in the sialoadenectomy group for analysis. Future research should examine these variables, as alterations in the oral microbiome may impact the intestinal microbiome, adding to evidence of a “oral–gut axis”.

The plausible mechanisms by which reduced saliva impacts intestinal conditions may involve: (1) oral microbiome dysbiosis resulting from decreased saliva, subsequently leading to dysbiosis of the intestinal microbiome; and (2) reductions in salivary EGF and/or VEGF lowering intestinal levels of these growth factors, resulting in structural changes in the intestine, including alterations in villus morphology, goblet cell density, and Paneth cell structure (Figure 12).

## 4. Materials and Methods

### 4.1. Animal Models

All animal experiments were approved by the ethical committee of our institution in accordance with the guidelines on animal experiments of Tsurumi University School of Dental Medicine in July 2022 (approval number: 22P022).

We used 6-week-old male C57BL/6J mice (CLEA Japan Inc., Tokyo, Japan) with an average body weight of 21.2 g in this study. They were maintained under specific pathogen-free conditions at a constant room temperature with a 12/12 h light/dark cycle throughout the experiment, and were provided access to paste food chow (CE-2; SLC Co., Shizuoka, Japan; CE-2: water = 1:2) and sterilized water ad libitum.

In this study, humane endpoints were defined to prevent undue distress in mice. Animals were monitored daily for general health indicators such as weight loss (exceeding 20% of baseline), lack of grooming, hunched posture, and changes in behavior including reduced mobility or abnormal feeding patterns. Especially after surgical treatment, we frequently monitored them to detect adverse signs. We decided that if signs of severe distress or unresponsiveness to treatment were observed, humane euthanasia would be performed; however, any unexpected adverse event did not occur. All monitoring was conducted by personnel trained to recognize these clinical signs, ensuring rapid and compassionate intervention when necessary.

### 4.2. Experimental Procedure

After acclimation for 1 week upon arrival at our institution, the mice were randomly divided into sialoadenectomy, sham, and control groups. The sialoadenectomy group underwent surgical removal of the major salivary glands, the sham group received only an incision and elevation of the flap under general anesthesia, and the control group received neither anesthesia nor surgical intervention. General anesthesia was administered intraperitoneally using a combination of medetomidine hydrochloride (Nippon Zenyaku Kogyo Co., Ltd., Fukushima, Japan), midazolam (Astellas Pharma Inc., Tokyo, Japan), and butorphanol (Meiji Seika Pharma Co., Ltd., Tokyo, Japan) [57]. Once fully anesthetized, the mice were gently fixed in the supine position with rubber bands, and the surgical area was sterilized. For the sialoadenectomy group, an incision was made from the anterior neck to the upper thorax area, and flaps were elevated at the subcutaneous layer toward the submandibular area bilaterally. The submandibular and sublingual glands were exposed and resected, followed by parotid gland resection. The incision was then closed with 4-0 silk sutures [58,59]. The same incision and flap elevation as in the sialoadenectomy group was performed in the sham group, but no glands were resected, and the incision was closed with 4-0 silk sutures. For the control, neither general anesthesia nor any incision was performed. All three groups were analyzed 3, 6, and 12 months after surgery.

Based on the 3Rs principle (replacement, reduction, and refinement) for ethical and humane animal use, as well as an assessment of expected outcomes and statistical requirements, the sample sizes in this study were determined as outlined in Table 6.

### 4.3. Measurement of Body Weight and Salivary Secretion

Body weight and salivary secretion were measured monthly during the experimental period. The mice were weighed using an electronic balance scale for animals (Entris II; Sartorius, Göttingen, Germany). Then, they were anesthetized as previously described [57] and positioned supine without fixation. Pilocarpine hydrochloride (0.1 mg/kg; Sampiro Ophthalmic Solution; Santen Pharmaceutical Co., Ltd., Osaka, Japan) was injected intraperitoneally to stimulate salivation. Extra-Thick Blot Filter Paper (Bio-Rad, Hercules, CA, USA) was cut to an appropriate size and placed in the oral cavity for 30 min to collect the secreted saliva [60,61]. The difference in the filter paper’s weight before and after saliva absorption was recorded and used to calculate the amount of secreted saliva. Medetomidine hydrochloride antagonist (10 mg/kg; Nippon Zenyaku Kogyo Co., Ltd.) was injected intraperitoneally to expedite anesthesia recovery. The data of body weight and salivary secretion were statistically analyzed for all three groups.

### 4.4. Preparation of Fecal and Intestinal Samples

At 3, 6, and 12 months after surgery, the mice in each group were euthanized by cervical dislocation under full general anesthesia to collect fecal and intestinal samples [62]. The digestive tract, including the esophagus, stomach, duodenum, jejunum, ileum, cecum, and colon, was harvested using the Swiss-rolling technique [63,64]. Briefly, the abdominal skin was incised, and the whole intestines were carefully extracted from the distal end of the rectum/anus junction to the cecal junction. The colon was longitudinally incised for the immediate collection of fecal samples, which were instantly frozen and stored at −20 °C until further analysis [62]. The remaining digestive tract was gently extracted, cleaned of mesenteric connective and fat tissue, longitudinally incised along the mesenteric line, and thoroughly washed with phosphate-buffered saline (PBS). Then, the incised digestive tract was laid flat on a rubber plate with the luminal side up and pinned before brief fixation with 10% formaldehyde (FUJIFILM Wako Pure Chemical Corporation, Osaka, Japan). The tract was gently rolled from the proximal end of the stomach to the cecum and colon, fixed with 10% formaldehyde, and stored at 4 °C for 48 h.

### 4.5. Histopathological Analysis—Villi Length, Thickness, and Area

After fixation, samples were immersed in 70% alcohol for 48 h and embedded in paraffin (SAKURA Tissue-Tek, Sakura Finetek Japan, Tokyo, Japan). Tissue sections (5-µm thick) were mounted on silane-coated slides (New Silane III; Muto Pure Chemicals Co., Ltd., Tokyo, Japan), deparaffinized with xylene, and rehydrated using graded ethanol solutions (FUJIFILM Wako Pure Chemical Corporation). Following hematoxylin and eosin staining (Merck, Darmstadt, Germany), the sections were pathologically analyzed using a BX51 System Microscope, and images were captured using a DP70 digital microscope camera (Olympus, Tokyo, Japan) using Mosaic 2.4 software (Bio Tools K.K., Gunma, Japan). Villi length, thickness, and area were measured at 3, 6, and 12 months after surgery for each group in 10 randomly selected areas in the jejunum and ileum of each section using ImageJ software (National Cancer Institute, Bethesda, MD, USA).

### 4.6. Diastase Periodic Acid–Schiff (d-PAS) Staining and the Number of Goblet Cells per Unit Area

We used d-PAS staining to assess glycogen content in the digestive tract. Briefly, the sections were deparaffinized and rehydrated with tap water, followed by soaking in water with small amount of saliva at 37 °C for 30 min. Next, the sections were treated for 5 min treatment with 0.5% periodic acid solution (Muto Pure Chemicals Co., Ltd.), followed by 15 min with Schiff reagent (Sigma-Aldrich Corp., St. Louis, MO, USA) at room temperature in the dark. The sections were then soaked three times in sulfurous acid water (Muto Pure Chemicals Co., Ltd.) for 3 min, followed by Meyer hematoxylin (Sigma-Aldrich Corp.) staining for 2 min. After rinsing for 10 min in water, the sections were dehydrated in ethanol, cleared with xylene, and sealed with a coverslip [65,66].

We captured photos of the samples using the previously mentioned microscope setting and three authors (K.M., Mi.M., and Mo.M) manually counted the number of d-PAS-positive goblet cells to ensure consistency and accuracy in the cell quantification. Then, the number of goblet cells per 10,000 μm^2^ was calculated based on previously measured areas of villi.

### 4.7. Immunohistochemistry and Semi-Quantitative Analysis

After the sections were deparaffinized with xylene and rehydrated through descending ethanol concentrations, we used Immunosaver (Nisshin EM, Tokyo, Japan) for antigen retrieval according to the manufacturer’s protocol. Briefly, the sections were incubated in Immunosaver (1:200 dilution in tap water) for 40 min at 95 °C and transferred to tap water for incubation for 10 min at room temperature. Subsequently, the slides were treated for 30 min at room temperature with 3% hydrogen peroxide in methanol (both FUJIFILM Wako Pure Chemical Corporation) to inactivate endogenous peroxidase and 20% normal goat serum (Nichirei Biosciences Inc., Tokyo, Japan) for 30 min at room temperature. Then, the sections were incubated overnight at 4 °C with the following primary antibodies: rabbit polyclonal anti-epidermal growth factor (EGF) antibody (1:400; Bioss, Woburn, MA, USA), anti-vascular endothelial growth factor (VEGF) antibody (1:300; Bioss), and anti-poly(ADP-ribose) polymerase (PARP) antibody (1:400; Abcam, Cambridge, UK).

The antibodies were diluted with PBS (pH 7.4) containing 1% bovine serum albumin (Sigma-Aldrich Corp.). After several PBS washes, Histofine Simple Stain MAX PO (Rabbit) (Nichirei Biosciences Inc.) and DAB (Vector Laboratories, Burlingame, CA, USA) were used to visualize the bound antibody according to the manufacturer’s protocol. The sections were counterstained with hematoxylin and mounted. For negative controls, sections were processed without exposure to the primary antibody.

For semiquantitative analysis, we used ImageJ software with the Color Deconvolution plugin [67]. Shortly, images were captured under consistent settings, opened in ImageJ, and the “H DAB” option in the Color Deconvolution plug-in was applied to isolate the DAB channel. The DAB channel image was converted to 8-bit grayscale and a set threshold to define positively stained areas. Using the “Analyze Particles” function, we calculated the percentage of the stained area and integrated density (area × mean gray value) for each image. Staining intensity was averaged across multiple fields per sample, allowing for a semi-quantitative assessment of antigen expression.

### 4.8. DNA Extraction and 16S rRNA Gene Amplicon Sequencing

#### 4.8.1. DNA Extraction from Fecal Samples

Total genomic DNA was extracted from fecal samples using the ISOSPIN Fecal DNA Kit (Nippon Gene Co., Ltd., Tokyo, Japan) according to the manufacturer’s protocol. To assess the quality and quantity of extracted DNA, we used a spectrophotometer (NanoDrop One; Thermo Fisher Scientific Inc., Waltham, MA, USA) and confirmed DNA integrity using 1% agarose gel electrophoresis.

#### 4.8.2. 16S rRNA Gene Amplification and Library Preparation

This step was outsourced to Genome-Lead Co., Ltd. (Takamatsu, Kagawa, Japan). To characterize the microbial communities in the samples, we targeted the V3–V4 hypervariable regions of the bacterial 16S rRNA gene. A composite pair of primers with unique 17- or 21-base adapters was designed, enabling sample multiplexing and compatibility with Illumina sequencing. The used primer set was as follows:
FTCGTCGGCAGCGTCAGATGTGTATAAGAGACAGNCCTACGGGNGGCWGCAGRGTCTCGTGGGCTCGGAGATGTGTATAAGAGACAGNGACTACHVGGGTATCTAATCC

The initial PCR was performed to amplify the V3–V4 regions from each sample. Following this, a second round of PCR was conducted using primers containing both adapter and index sequences to identify samples comparing with pooled library format [17].

#### 4.8.3. Sequencing on Illumina MiSeq Platform

This step was also outsourced to Genome-Lead Co., Ltd. (Takamatsu, Kagawa, Japan), where a 2 × 301 bp paired-end sequencing run was conducted on the Illumina MiSeq platform (Illumina Inc., San Diego, CA, USA).

#### 4.8.4. Bioinformatics Analysis

After sequencing was completed, the raw data were sent to us, and were processed and analyzed by α-diversity, β-diversity, linear discriminant analysis (LDA) effect size (LEfSe) algorithm, and heatmap analysis using bioinformatics software QIIME2 (v2021.4). Taxonomic assignment was performed against reference database (Silva 128 SEPP reference database, MD5: 7879792a6f42c5325531de9866f5c4de) to profile the microbial communities.

### 4.9. Statistical Analysis

Data collected from the three experimental groups, including measurements of body weight, saliva secretion, villi morphology, d-PAS-positive goblet cell counts, semi-quantitative immunohistochemistry (IHC) analysis, and microbiome composition ratios, were analyzed using EZR software ver 1.62 (Saitama Medical Center, Jichi Medical University, Saitama, Japan), a graphical user interface for R (The R Foundation for Statistical Computing, Vienna, Austria). This modified version of R Commander is designed to add statistical functions frequently used in biostatistics [68]. Specifically, one-way analysis of variance and multiple comparisons using Bonferroni correction were performed. *p*-values < 0.05 were considered statistically significant.

For microbiome analysis, the within-subject α-diversity of bacterial communities was assessed using the Shannon index and the observed number of operational taxonomic units. These indices were compared among the three groups using the Kruskal–Wallis test. Between-subject β-diversity was evaluated based on Bray–Curtis dissimilarity and unweighted and weighted UniFrac distance metrics. We performed principal coordinate analysis to visualize the global differences in microbiome structure in the UniFrac analysis. The significance of compositional differences between groups was assessed by permutational multivariate analysis of variance. QIIME2 v2021.4 was utilized for these analyses [17,62,69,70].

We used the linear discriminant analysis (LDA) effect size (LEfSe) algorithm to identify differentially abundant genera among the three groups. All analyses were performed with LEfSe’s α parameter set to <0.05 for pairwise tests and the threshold of the logarithmic score for LDA set to >4.0. Following model construction, the variable importance in projection of each level (from phylum to species) was calculated, and the candidate genera were selected to reflect differences between the groups [71,72]. *p*-values < 0.05 were considered statistically significant. We performed heatmap analysis to detect the differential abundance of microbial taxa across the three groups at various taxonomic hierarchies, from phylum to species [73,74].

## 5. Conclusions

In conclusion, our study highlights the complex interplay between saliva secretion, intestinal histopathology, and microbiome composition. Reduced saliva secretion due to sialoadenectomy significantly impacted villi morphology, goblet cell density, Paneth cell structure, and microbiome composition in the short term, and these effects diminished over time. The unexpected favorable outcomes in the sham group suggest that minor surgical interventions stimulate immune responses that enhance overall intestinal health. Our findings underscore the need for further research to fully elucidate the underlying mechanisms and explore their relevance to human health.

## Figures and Tables

**Figure 1 ijms-25-12455-f001:**
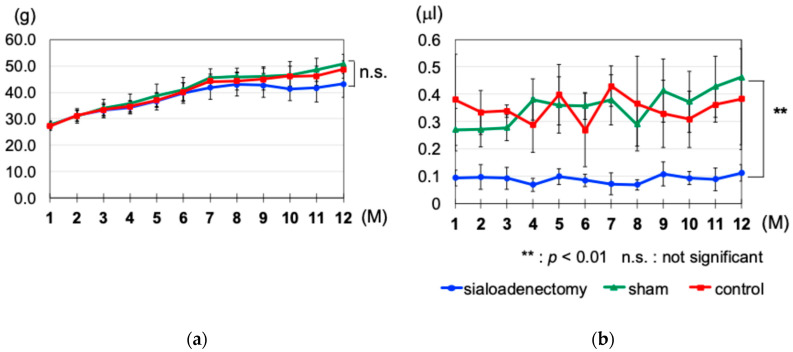
Changes in body weight and saliva secretion during the experimental period. (**a**) Changes in body weight during the experimental period were not significantly different among the three groups. The sialoadenectomy group showed a slower increase in body weight 6 months after surgery, but this did not reach statistical significance. (**b**) Changes in saliva secretion during the experimental period. The amount of saliva secreted in the sialoadenectomy group was significantly lower than the other two groups during the experimental period. ** *p* < 0.01; n.s., not significant.

**Figure 2 ijms-25-12455-f002:**
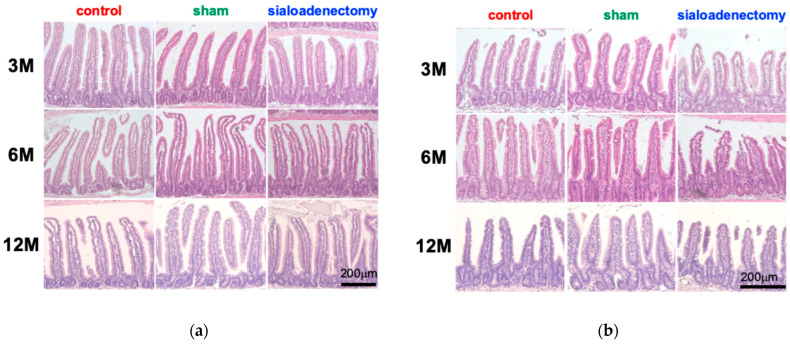
Morphological changes in intestinal villi. (**a**) Representative hematoxylin and eosin-stained images of jejunal villi from each group at 3, 6, and 12 months after surgery. (**b**) Representative hematoxylin and eosin-stained images of ileal villi from each group at 3, 6, and 12 months after surgery. Scale bar: 200 μm.

**Figure 3 ijms-25-12455-f003:**
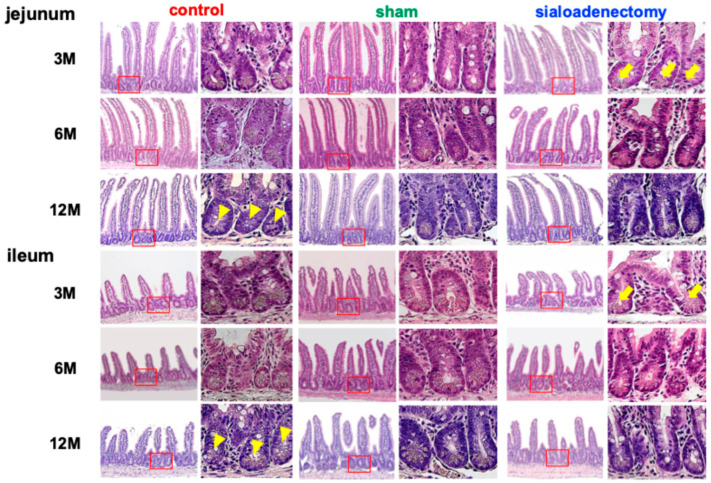
Histopathological changes in Paneth cells. Representative hematoxylin and eosin-stained Paneth cells at 3, 6, and 12 months after surgery for each group. The right photo in each column is a magnified view of the red box in the corresponding left photo. The Paneth cells in the sialoadenectomy group showed deformation (arrows) that recovered over time. The control group showed slight deformation 12 months after surgery (arrowheads).

**Figure 4 ijms-25-12455-f004:**
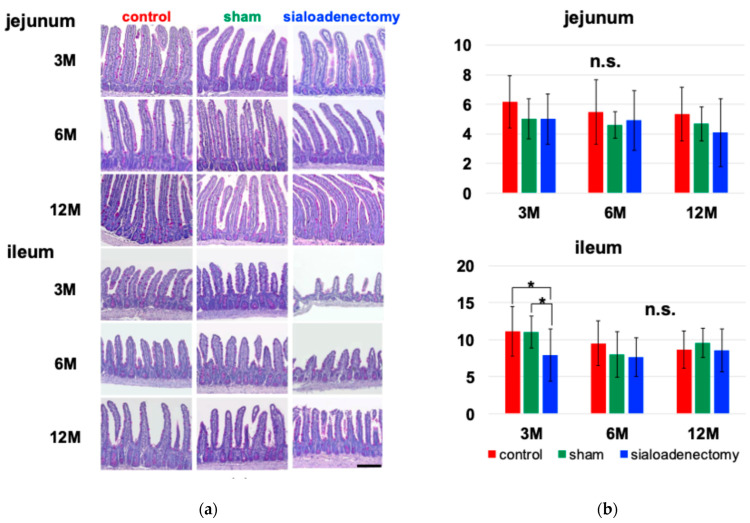
Analysis of sialoadenectomy effects on the number of goblet cells using d-PAS staining. (**a**) d-PAS staining of the jejunum (upper) and ileum (lower) at 3, 6, and 12 months after surgery. Scale bar: 200 μm. (**b**) The d-PAS-positive goblet cell density per unit area in the ileum was significantly decreased in the sialoadenectomy group at 3 months after surgery compared to the sham and control groups, but this difference was not significant at 6 or 12 months after surgery. In contrast, goblet cell density in the jejunum showed no significant differences among the three groups throughout the experimental period. * *p* < 0.05, n.s., not significant.

**Figure 5 ijms-25-12455-f005:**
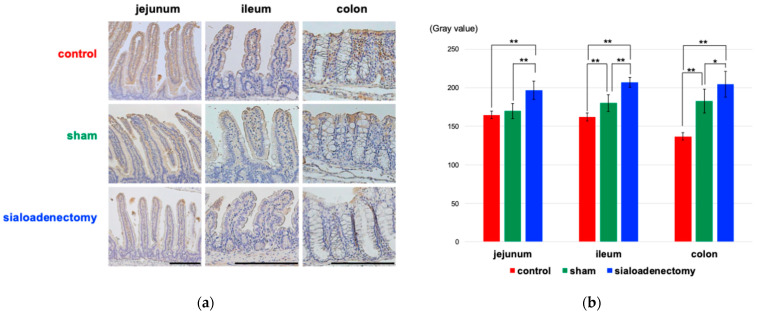
Immunohistochemical analysis of EGF expression in the sialoadenectomy, sham, and control groups. Representative results for each group are shown. (**a**) EGF expression was detected at the epithelial surface of the jejunum, ileum, and colon in the control (upper row) and sham (middle row) group 3 months after surgery and was relatively weak in the sialoadenectomy group (lower row). Scale bar: 200 μm. (**b**) Semi-quantitative analysis following conversion to grayscale. The gray values in the sialoadenectomy group were significantly higher (equivalent to decreased EGF expression) compared with the other two groups in the jejunum, ileum, and colon. Values are expressed as mean ± SEM. * *p* < 0.05, ** *p* < 0.01.

**Figure 6 ijms-25-12455-f006:**
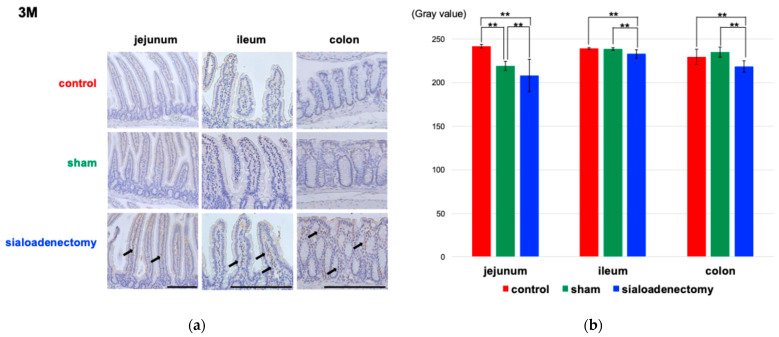
Immunohistochemical analysis of anti-PARP-1 antibody at 3 months after surgery. (**a**) The sialoadenectomy group (lower row) showed significantly more PARP-1-positive cells compared with the control (upper row) and sham (middle row) groups in the jejunum, ileum, and colon. Arrows: PARP-1-positive cells. Scale bar: 200 μm. (**b**) The gray values were measured using ImageJ (1.54 f) software and compared among the three groups. At 3 months after surgery, the sialoadenectomy group showed significantly higher PARP-1 positivity in the jejunum, ileum, and colon compared with the control and sham groups. Values are expressed as mean ± SEM. ** *p* < 0.01.

**Figure 7 ijms-25-12455-f007:**
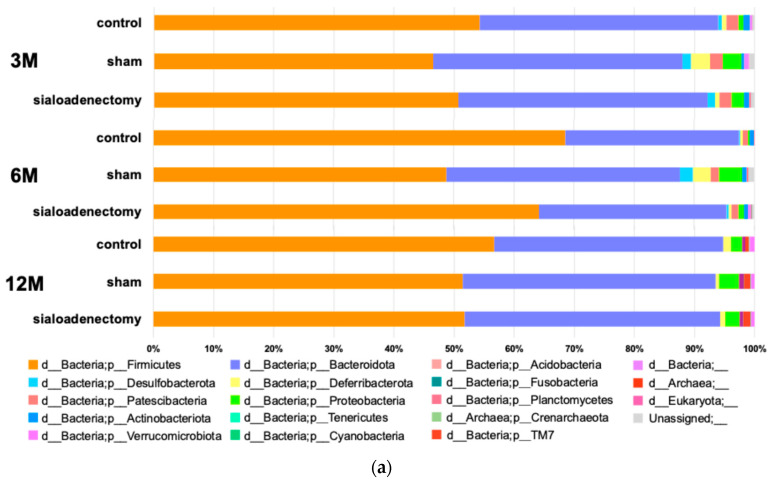
Microbiome composition across the three groups at 3, 6, and 12 months after surgery. (**a**) Microbiome composition at the phylum level. No specific phylum level microbiome was consistently detected throughout the experimental period. (**b**) Microbiome composition at the genus level. Significant differences in relative abundance were observed among the groups at 3, 6, and 12 months after surgery. The genera shown in the figure exhibited statistically significant variations in mean relative abundance, reflecting distinct microbiome characteristics within each group. *Lactobacillus* was observed in all three groups at 3 and 6 months after surgery, with significant differences across the three groups. However, 12 months after surgery, *Lactobacillus* was the most abundant microbiota in all three groups, and there was no significant difference across the groups.

**Figure 8 ijms-25-12455-f008:**
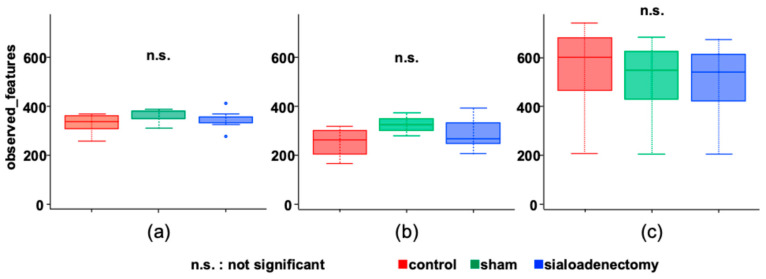
Within-subject α-diversity. There was no significant difference among the three groups at (**a**) 3, (**b**) 6, and (**c**) 12 months after surgery.

**Figure 9 ijms-25-12455-f009:**
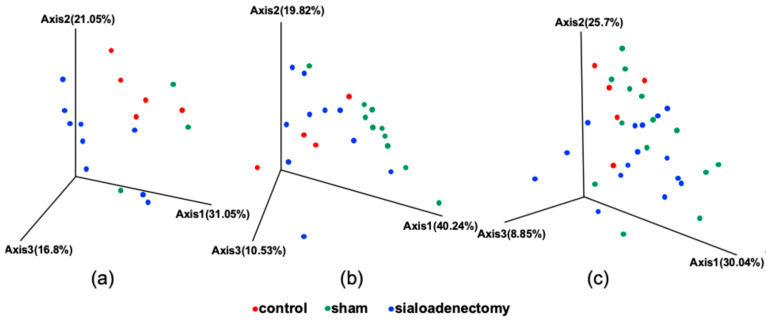
β-diversity of microbial communities at (**a**) 3, (**b**) 6, and (**c**) 12 months after surgery. Significant differences were observed between control and sialoadenectomy groups at 3 months after surgery and between the sham group and both control and sialoadenectomy groups at 6 months after surgery (*p* < 0.05).

**Figure 10 ijms-25-12455-f010:**
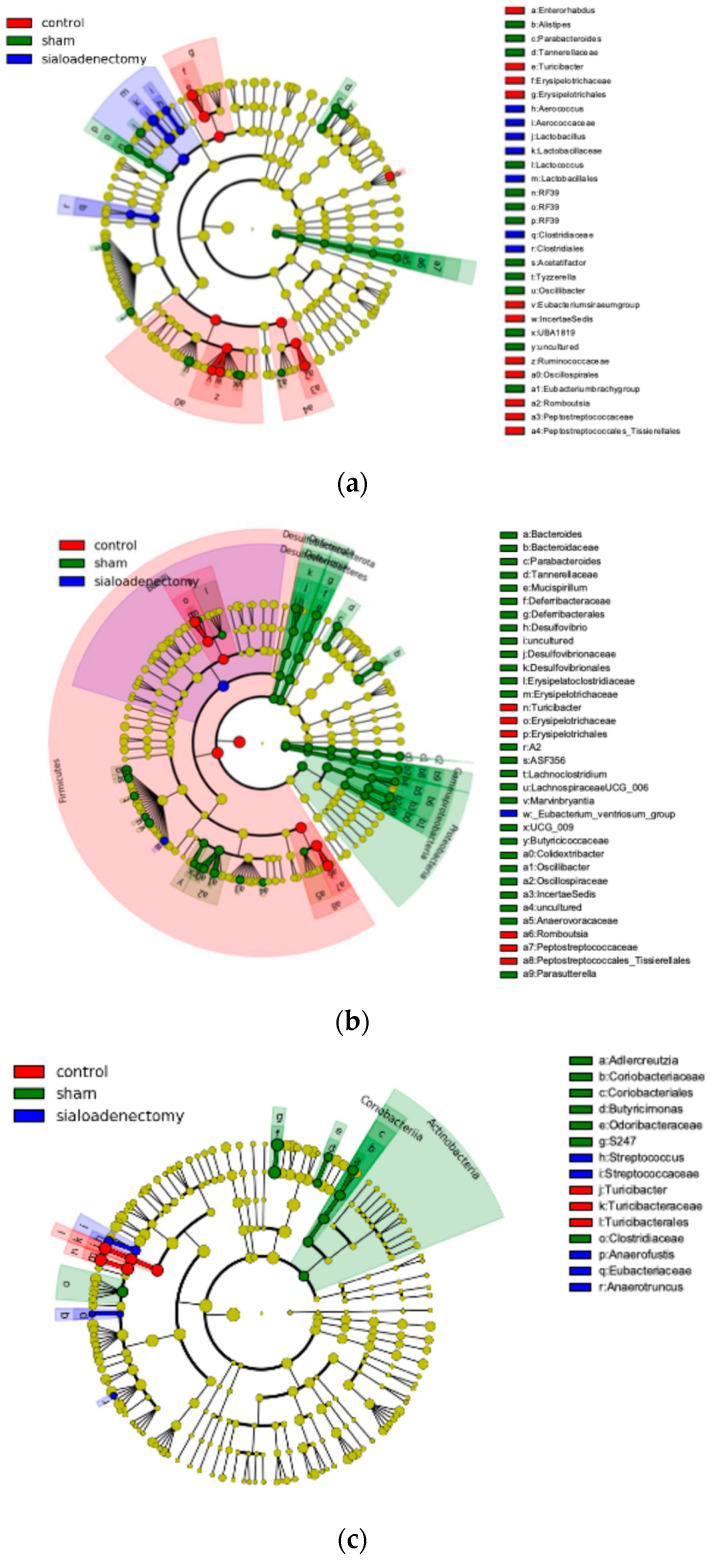
Cladograms plotted from the LEfSe analysis. Taxonomic changes in intestinal microbiota across the three groups at (**a**) 3, (**b**) 6, and (**c**) 12 months after surgery.

**Figure 11 ijms-25-12455-f011:**
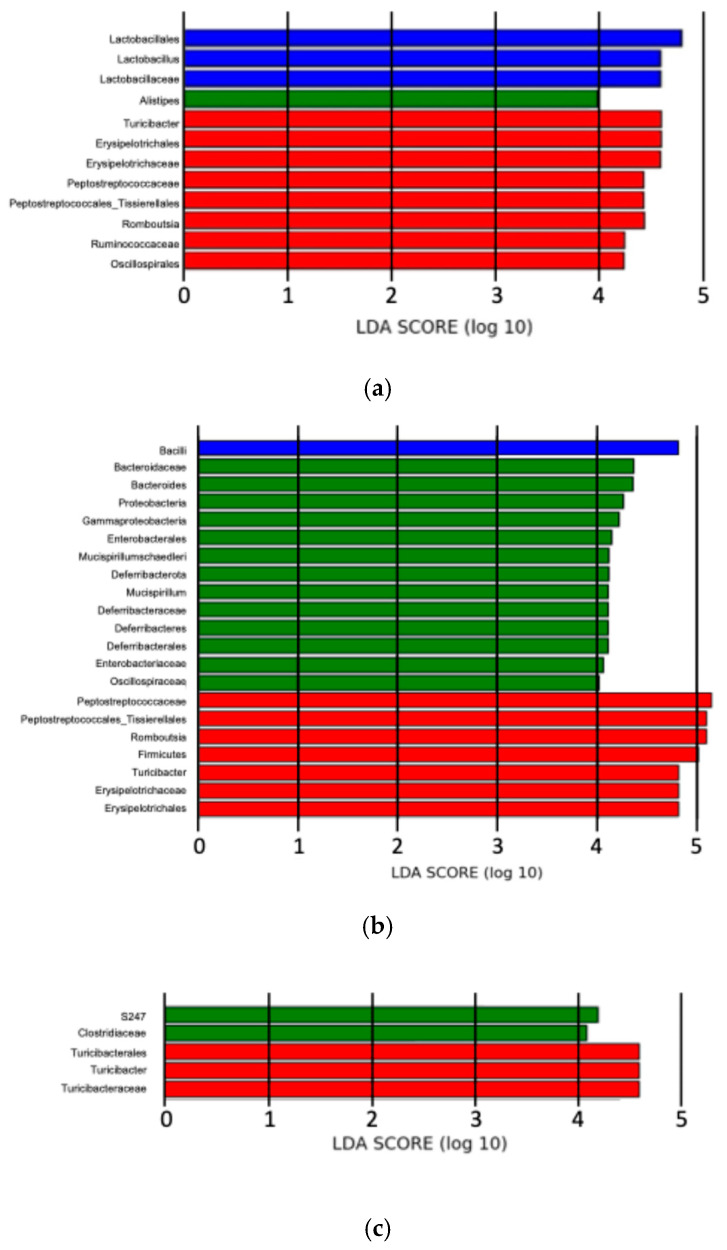
Differentially abundant microbiota across the three groups identified using LEfSe. Panels represent results at (**a**) 3, (**b**) 6, and (**c**) 12 months after surgery. Log-transformed LDA scores are plotted on the x-axis. LDA scores > 4 were considered statistically significant. Bar length indicates the relative influence of each microbiota.

**Figure 12 ijms-25-12455-f012:**
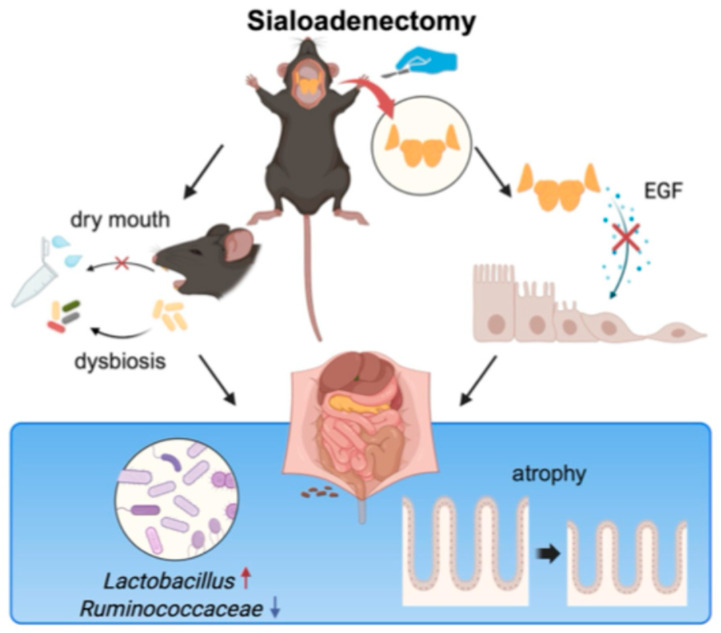
Possible and plausible mechanisms of the effect of sialoadenectomy on the intestinal microbiome. Sialoadenectomy induces xerostomia (dry mouth), which is associated with oral microbiome dysbiosis, subsequently affecting the intestinal microbiome. Sialoadenectomy also reduces epidermal growth factor (EGF) secretion, which may directly alter the intestinal epithelium, further contributing to intestinal microbiome changes.

**Table 1 ijms-25-12455-t001:** Length (μm) of jejunal and ileal villi of three groups. *: significant difference compared to the other two groups. *p* < 0.05.

		Control	Sham	Sialoadenectomy
jejunum	3 M	375.2 ± 83.8	348.5 ± 105.2	292.0 ± 69.5 *
	6 M	347.5 ± 74.2	348.3 ± 81.3	331.6 ± 76.6
	12 M	326.8 ± 74.1	381.5 ± 73.5 *	311.2 ± 66.8
ileum	3 M	221.1 ± 33.9	200.1 ± 40.7	184.5 ± 42.7 *
	6 M	241.1 ± 39.8	241.2 ± 45.0	200.7 ± 29.6 *
	12 M	199.7 ± 37.2	232.1 ± 30.6 *	198.1 ± 34.6

**Table 2 ijms-25-12455-t002:** Thickness (μm) of jejunal and ileal villi of three groups. Significant difference compared to the other two groups. *: *p* < 0.05, **: *p* < 0.01.

		Control	Sham	Sialoadenectomy
jejunum	3 M	74.5 ± 11.7 *	62.1 ± 7.1	63.6 ± 9.2
	6 M	67.7 ± 12.9	71.1 ± 9.6	67.1 ± 10.8
	12 M	69.5 ± 6.9	63.7 ± 7.8	76.2 ± 7.3 **
ileum	3 M	54.5 ± 7.5	56.4 ± 5.7	52.2 ± 4.4
	6 M	54.3 ± 7.8	68.9 ± 16.0 *	62.6 ± 7.2
	12 M	52.6 ± 5.0	54.7 ± 8.2	58.6 ± 9.2

**Table 3 ijms-25-12455-t003:** Area (μm^2^) of jejunal and ileal villi of three groups. **: significant difference compared to the other two groups. *p* < 0.01.

		Control	Sham	Sialoadenectomy
jejunum	3 M	21,296.3 ± 4577.8	17,867.3 ± 4082.4	14,530.35 ± 4842.8 **
	6 M	17,841.8 ± 4727.2	19,743.7 ± 4603.0	19,742.2 ± 5813.5
	12 M	17,692.5 ± 4154.8	17,922.7 ± 2396.2	21,746.5 ± 5106.7
ileum	3 M	10,458.0 ± 2197.5	8776.4 ± 1964.5	6833.5 ± 2178.4
	6 M	7755.5 ± 1401.0	9854.7 ± 2972.0	8720.9 ± 1086.82
	12 M	8874.1 ± 1053.3	9544.5 ± 1195.3	9641.5 ± 1785.2

**Table 4 ijms-25-12455-t004:** The number of goblet cells observed per villi. *: significant difference compared to the other two groups. *p* < 0.05.

		Control	Sham	Sialoadenectomy
jejunum	3 M	11.8 ± 4.3 *	7.6 ± 2.7	7.9 ± 3.3
	6 M	8.4 ± 2.9	7.8 ± 2.2	8.4 ± 2.0
	12 M	9.0 ± 3.1	9.1 ± 3.6	8.9 ± 2.4
ileum	3 M	9.8 ± 2.7 *	7.1 ± 2.4	6.2 ± 2.4
	6 M	7.7 ± 2.1 *	6.5 ± 2.2	6.6 ± 2.0
	12 M	8.1 ± 2.1	8.1 ± 2.0	8.3 ± 2.1

**Table 5 ijms-25-12455-t005:** Summary of commonly upregulated microbiomes identified in composition ratio, LEfSe, and heatmap analysis by group and time point.

Group	Time Point	Upregulated Microbiome
Sialoadenectomy	3 M	o_*Lactobacillales*; f_*Lactobacillaceae*; g_*Lactobacillus*
Control	3 M	f_*Ruminococcaceae*; g_*Incertae_Sedis*
	3 M	o_*Peptostreptococcales-Tissierellales*; f_*Peptostreptococcaceae*
	3 M	o_*Oscillospirales*; f_*Ruminococcaceae*; g_*Romboutsia*,
	6 M	f_*Peptostreptococcaceae*; g_*Romboutsia*
	6 M	o_*Erysipelotrichales*; f_*Erysipelotrichaceae*; g_*Turicibacter*

**Table 6 ijms-25-12455-t006:** The number of mice which are experienced in this study.

	3 M	6 M	12 M
Sialoadenectomy Group	10	10	15
Sham Group	9	10	14
Control Group	5	5	5

## Data Availability

16S rRNA gene amplicon sequence data were registered to Sequence Read Arcive (SRA) on NCBI website (PRJNA1173082).

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
