# Peer review of "Impact of Reduced Saliva Production on Intestinal Integrity and Microbiome Alterations: A Sialoadenectomy Mouse Model Study"

_ijms, 2024, doi:10.3390/ijms252212455_

Round 1

Reviewer 1 Report

Comments and Suggestions for Authors

The authors sought to evaluate the impact of reduced saliva production on gastrointestinal integrity and changes in the microbiome. Although the study is interesting, some questions need to be resolved before its publication.

First, although the authors observed some gastrointestinal and microbiome changes in animals undergoing sialadenectomy, it is important to be cautious when writing and discussing the results, so as not to sound overly speculative. It is necessary to expand the discussion, based on results from previous studies and justifying, in light of the current literature, how surgical removal of the salivary glands can cause disturbances in the microbiota and morphology of the gastrointestinal tract. The results are very interesting, but the discussion is superficial. I suggest rewriting it, emphasizing the results obtained.

What previous studies served as a basis for performing the surgical procedure? Although the protocol is well described, there is no reference to the method used. Was this model developed by you?

Why was saliva quantification performed by administering pilocarpine, a drug that stimulates salivary secretion? Why didn't the authors perform the experiment in a way that verified saliva secretion without any drug stimulation?

Why wasn't the duodenal region of the animals evaluated histologically?

In the title of the article, the authors state that this is an evaluation of the gastrointestinal tract, but at no point is there any mention of an analysis of the impact on the stomach, for example. Wouldn't it be interesting to change the title to include only intestinal changes?

The authors adequately mention how the photomicrographs were captured to evaluate the length of the villi, but it is not clear which tool was used to quantify the height. How was this quantification performed? Why was only the length and not the thickness of the villi evaluated?

The density of goblet cells was also not considered; I believe that this parameter would also be interesting to include in the study.

The authors state that glycogen content was assessed using special stains such as PAS, but again, it is not clear how this was quantified, making it difficult to reproduce this methodology in future studies that seek to base themselves on the data presented.

The authors mention that the immunohistochemical results were assessed semiquantitatively. How was this done? Please describe them appropriately.

I did not find the graphs and images of the immunohistochemical analyses for VEGF in the manuscript. Why were they not included?

I would expect the authors to better correlate the data on VEGF immunoexpression with EGF. Do you not find it contradictory that EGF expression increased in animals submitted to sialadenectomy compared to the control group? In my opinion, it is contradictory to observe a greater expression of EGF and, at the same time, atrophy/decrease in the length of the villi. Please explain this in detail.

Finally, the assessment was restricted to the intestine, without involving the entire gastrointestinal tract, which includes other structures and accessory organs. I therefore suggest modifying several passages in which the authors refer to the results, restricting them to "intestinal" changes.

Author Response

Thank you very much for your valuable and insightful comments, which have greatly contributed to enhancing the quality of our manuscript. We have thoroughly and carefully revised the manuscript in accordance with your recommendations, with all changes highlighted in yellow. I hope these modifications meet the standards you have requested.

The authors sought to evaluate the impact of reduced saliva production on gastrointestinal integrity and changes in the microbiome. Although the study is interesting, some questions need to be resolved before its publication.

  1. First, although the authors observed some gastrointestinal and microbiome changes in animals undergoing sialadenectomy, it is important to be cautious when writing and discussing the results, so as not to sound overly speculative. It is necessary to expand the discussion, based on results from previous studies and justifying, in light of the current literature, how surgical removal of the salivary glands can cause disturbances in the microbiota and morphology of the gastrointestinal tract. The results are very interesting, but the discussion is superficial. I suggest rewriting it, emphasizing the results obtained.

Reply: According to you your advice, we thoroughly reviewed our manuscript and re-write whole discussion part.

  1. What previous studies served as a basis for performing the surgical procedure? Although the protocol is well described, there is no reference to the method used. Was this model developed by you?

Reply: Our methodology is a mix of elements from two previously published studies. Although prior literatures had documented the use of sialoadenectomy, these reports typically provide only brief mention, stating that 'sialoadenectomy was performed' without procedural specifics. Consequently, comprehensive search on PubMed was conducted, yielding two articles that thoroughly outline the procedural details.

  1. Zubeidat, K.; Saba, Y.; Barel, O.; Shoukair, F.L.; Hovav, A.H. Protocol for Parotidectomy and Saliva Analysis in Mice. STAR Protoc. 2022, 3, 101048, doi:10.1016/j.xpro.2021.101048.
  2. Jonjic, S. Surgical Removal of Mouse Salivary Glands. Curr. Protoc. Immunol. 2001, 43, 1.11.1-1.11.4, doi:10.1002/0471142735.IM0111S43.

Therefore, we added two manuscripts for the references (59 and 60) for the sialoadenectomy procedures (Line 504).

  1. Why was saliva quantification performed by administering pilocarpine, a drug that stimulates salivary secretion? Why didn't the authors perform the experiment in a way that verified saliva secretion without any drug stimulation?

Reply: Saliva quantification was conducted following established protocols (Ref 61). Pilocarpine-induced saliva secretion does not accurately and directly represent the physiological conditions after sialoadenectomy, thus, we initially attempted to measure its secretion without pilocarpine using micropipette. However, saliva production in sialoadenectomy group was minimal and highly variable, raising concerns about the feasibility and reliability of results. Consequently, we discontinued natural saliva measurement and opted to use pilocarpine to stimulate secretion.

  1. Why wasn't the duodenal region of the animals evaluated histologically?

In the title of the article, the authors state that this is an evaluation of the gastrointestinal tract, but at no point is there any mention of an analysis of the impact on the stomach, for example. Wouldn't it be interesting to change the title to include only intestinal changes?

Reply: Unfortunately, due to limitations in sample quality of esophageal, gastric, and duodenal regions, we were unable to conduct a thorough histopathological analysis especially for upper gastrointestinal levels. While some samples met the criteria for basic histological assessment, others were insufficient for comprehensive examination.

We had intended to evaluate the duodenal region, as it contains Brunner’s glands, which produce EGF. Fortunately, the quality of lower digestive tract samples was sufficient for histopathological evaluation.

Additionally, we have carefully reviewed and revised our manuscript to reflect this scope more accurately. To precisely represent our findings, including in the title, we replaced several instances of “gastrointestinal” with “intestinal” throughout the manuscript.

  1. The authors adequately mention how the photomicrographs were captured to evaluate the length of the villi, but it is not clear which tool was used to quantify the height. How was this quantification performed? Why was only the length and not the thickness of the villi evaluated?
  2. The density of goblet cells was also not considered; I believe that this parameter would also be interesting to include in the study.

Reply. Regarding methodological approach, we used “ImageJ software (National Cancer Institute, Bethesda, MD, USA)” to evaluate the length of the villi. To quantify the number of goblet cells, we counted d-PAS positive cells manually by three authors (K.M., Mi.M., and Mo.M) to ensure the consistency and accuracy in cell counts.

Initially, we did not notice the importance to evaluate the thickness of villi. Following your suggestion, we added the analysis of villi length and thickness. Additionally, we recognized the importance of goblet cell density in this context, we quantified goblet cell density within the villi per unit area (10,000 μm²), derived from calculated area using ImageJ. We have updated the Methods and Results sections accordingly, as well as provided an expanded discussion part. (Table 2 and 3).  

  1. The authors state that glycogen content was assessed using special stains such as PAS, but again, it is not clear how this was quantified, making it difficult to reproduce this methodology in future studies that seek to base themselves on the data presented.

Reply: In the section of “4.6 Diastase Periodic Acid-Schiff (d-PAS) staining”, we did not mention how we counted the number of d-PAS positive cells. We captured the photo of villi using previously mentioned microscope setting and counted the number of d-PAS positive goblet cells manually by three authors (K.M., Mi.M., and Mo.M) ensuring consistency and accuracy in cell quantification.

  1. The authors mention that the immunohistochemical results were assessed semiquantitatively. How was this done? Please describe them appropriately.

Reply: To assess the density of DAB staining of immunohistochemical analysis, we utilized ImageJ software with the color deconvolution plugin to quantify antigen expression, available as a free download from the National Cancer Institute (NCI).

The evaluation process followed established protocols as referenced in our manuscript on lines 562-564 (1st submission) which might be too short. Therefore, we added slightly more detailed protocol on lines 594-601 (revised submission).  

  1. I did not find the graphs and images of the immunohistochemical analyses for VEGF in the manuscript. Why were they not included?

Reply: The semi-quantitative analysis of DAB staining for VEGF expression did not reveal any statistically significant differences among the three groups, which was unexpected. As we hesitate to show insignificant results in the main text to occupy,

we have included the VEGF IHC photos and put them supplementary material (Figure S2) and omitted the corresponding semi-quantitative analysis. For similar reasons, we had omitted graphical data for the semi-quantitative analysis of EGF and PARP-1 at 6 and 12 months after surgery because they did not show any significant difference, neither. Thus, we briefly describe the VEGF results on Line 190-192 and Figure S2.

  1. I would expect the authors to better correlate the data on VEGF immunoexpression with EGF. Do you not find it contradictory that EGF expression increased in animals submitted to sialadenectomy compared to the control group? In my opinion, it is contradictory to observe a greater expression of EGF and, at the same time, atrophy/decrease in the length of the villi. Please explain this in detail.

Reply: In our study, EGF expression was significantly more positive in the control group at 3 months after surgery followed by insignificance after 6 and 12 months. VEGF expression remained consistently positive across all groups, with no significant differences observed throughout the entire experimental period. Considering that both EGF and VEGF are present in saliva, we had predicted that their expression levels in intestine would decrease in the sialoadenectomy group. However, only EGF expression showed a reduction in the sialoadenectomy group at 3 months, while VEGF levels did not exhibit any significant changes. Considering that the response of VEGF is usually earlier than EGF, thus, the VEGF decrease response had already been completed by 3 months after surgery, our first analysis time point. We mentioned the VEGF result more in detail and also commented in the discussion part as on lines 365-393.

  1. Finally, the assessment was restricted to the intestine, without involving the entire gastrointestinal tract, which includes other structures and accessory organs. I therefore suggest modifying several passages in which the authors refer to the results, restricting them to "intestinal" changes.

Reply: Thank you once again for your valuable guidance on terminology. We fully concur with your recommendations and have meticulously reviewed and revised the manuscript, replacing the term “gastrointestinal” with “intestinal” to more accurately reflect the scope of our findings. Several “gastrointestinal” remain in our manuscript considering the context and referred articles. 

Reviewer 2 Report

Comments and Suggestions for Authors

The aim  is investigate the effects of saliva reduction or sialoadenectomy on the gastrointestinal tract and intestinal microbiota. Consequently, we examined the impact of xerostomia on the gastrointestinal tract

The authors conclusion, the complex interplay between saliva secretion,intestinal histopathology, and microbiome composition. Reduced saliva secretion due to sialoadenectomy significantly impacted villi and Paneth cell structure and microbiome composition in the short term, and these effects diminished over time

Provide an accurate summary of the research objectives, animal species, strain and sex, key methods, principal findings, and study conclusions. Yes

Provide an accurate summary of the research objectives, animal species, strain and sex, key methods, principal findings, and study conclusions.Ok

Describe any interventions or steps taken in the experimental protocols to reduce pain, suffering and distress.  No

b. Report any expected or unexpected adverse events. No

c. Describe the humane endpoints established for the study, the signs that were monitored and the frequency of monitoring. No

It does not justify the number of animals per group.

A limitation of this study is that we were unable to evaluate saliva components or oral microbiome changes in the sialoadenectomy group due to insufficient saliva secretion

for analysis. The paper is excessively long, so it might be advisable to divide it."

The results confirm the study objectives and hypotheses, aligning with current theory and supporting findings from other relevant studies in the literature

Some of the figures do not look good.

Author Response

Thank you very much for your valuable and insightful comments, which have greatly contributed to enhancing the quality of our manuscript. We have thoroughly and carefully revised the manuscript in accordance with your recommendations, with all changes highlighted in light blue. I hope these modifications meet the standards you have requested.

The aim is investigate the effects of saliva reduction or sialoadenectomy on the gastrointestinal tract and intestinal microbiota. Consequently, we examined the impact of xerostomia on the gastrointestinal tract

The authors conclusion, the complex interplay between saliva secretion,intestinal histopathology, and microbiome composition. Reduced saliva secretion due to sialoadenectomy significantly impacted villi and Paneth cell structure and microbiome composition in the short term, and these effects diminished over time

  1. Provide an accurate summary of the research objectives, animal species, strain and sex, key methods, principal findings, and study conclusions. Yes

 Reply. Thank you very much for your kind consideration. However, another reviewers stated that the aims and scopes were beige in the first submission, therefore, we rewrote them almost drastically.

  1. Describe any interventions or steps taken in the experimental protocols to reduce pain, suffering and distress.  No

Reply. We performed the surgical procedure under general anesthesia and added cervical dislocation at the analysis point. We mentioned only general anesthesia, therefore, we state as follows on the line 530-531. 

At 3, 6, and 12 months after surgery, mice in each group were euthanized by cervical dislocation under full general anesthesia to collect fecal and intestinal samples[63].

  1. Report any expected or unexpected adverse events. No
  2. Describe the humane endpoints established for the study, the signs that were monitored and the frequency of monitoring. No

Reply. Replying to both 3rd and 4th comments, we described humane endpoints and how to treat them in at the “4.1 Animal Models” section on line 480-488.

  1. It does not justify the number of animals per group.

Reply. The sample size for the mouse groups was determined based on three key considerations: (1) statistical power and variability, (2) ethical standards in animal research, and (3) sufficiency for meaningful comparative analysis. To ensure adequate statistical power and account for anticipated effect sizes and variability based on prior studies, we calculated the minimum number of animals necessary to achieve statistical significance while adhering to the principle of ethical minimization. Balancing the need for sufficient sample sizes with ethical considerations, we determined that each group should contain more than 5 but does not need to exceed 10, in accordance with the 3Rs principle (Replacement, Reduction, and Refinement) to promote humane and ethical animal use. This range provided an optimal balance between statistical rigor and ethical responsibility. For animals assessed at 12 months after surgery, we used 15 mice in the sialoadenectomy group and 14 in the sham group, accounting for potential survival challenges associated with surgical intervention. This decision was made to decrease the risk of attrition affecting the statistical integrity of long-term outcomes.

To clarify our adherence to the 3Rs principle and the considerations of statistical power and variability in determining the sample sizes, we mentioned reason we chose the number based on the principle of 3Rs on the line 509-511.

  1. A limitation of this study is that we were unable to evaluate saliva components or oral microbiome changes in the sialoadenectomy group due to insufficient saliva secretion for analysis. The paper is excessively long, so it might be advisable to divide it."

Reply. Thank you for your valuable feedback. I understand your concern regarding the length of the manuscript. I have carefully reviewed the content and agree that the first submission could have benefited from a more concise discussion. However, the comprehensive nature of the manuscript is essential to fully address the scope of this research. As this is PhD thesis project for Ms. Maita, our first author, we aimed to include the full of her experimental contributions in a single manuscript to provide a cohesive and thorough presentation of her work. While it is possible to separate the histopathological and microbiome analyses into two distinct manuscripts, we believe the integration of both aspects is crucial for a holistic understanding of the study’s implications. Additionally, the revised version became longer than the previous one because of accommodating additional details requested by another reviewer. These revisions were necessary to address their feedback comprehensively.

We respectfully suggest that the length of the manuscript supports a detailed and integrated presentation that would be beneficial for readers interested in the interconnected findings of histopathology and microbiome analysis.

  1. The results confirm the study objectives and hypotheses, aligning with current theory and supporting findings from other relevant studies in the literature

Reply. Thank you very much for your comments, which made us encouraged.

  1. Some of the figures do not look good.

Reply. Another reviewer also highlighted similar concerns, particularly with Figure 11. Upon reviewing, we found that the resolution of certain figures may have been inadvertently reduced during the file preparation process, particularly when copying and pasting or inserting JPEG files into the Word document. To address this issue, we have now included high-resolution versions of all affected figures as supplementary material to ensure clarity and enhance the visual quality (Figure S4-S8). We hope this resolves any concerns regarding figure quality.

Reviewer 3 Report

Comments and Suggestions for Authors

·         Line 17: Clearly define “Xerostomia”.

·         Line 19: Define the study design here (e.g., This ? study investigated how…).

·         Line 23-29: Provide details about the statistical analysis used and significant p-value.

·         The introduction is very weak, fragmented, and not focused enough towards the research aim. The section should be qualified with more supporting evidence (e.g., Biores Open Access. 2017 Oct 1;6(1):123–132; J Oral Microbiol. 2022 Aug 11;14(1):2110194; Appl. Sci. 2020, 10, 6421). It is important to explore the mechanisms of microbial transmission, the impact of oral health on the gut microbiota alterations, and the potential role of dysbiosis in gut diseases both in vivo and in vitro/human studies. The main contribution or novelty needs a better explanation in the last paragraph.

·         Line 437: Clear justification why these groups were selected? Define clearly sham and control groups.

·         Line 466-469; Line 496-500; Line 546-563: The statistical  analysis should be in a separate section and described in sufficient details.

·         Line 532: Authors should clarify the rationale for using 16S rRNA Gene Amplicon Sequencing. This section should be described in more details.

·         It is unclear what test used to analyze gene expression (e.g, VEGF).

·         Line 121: Please make “Length of jejunal and ileal villi of three groups” the title of table 1 (move to line 120).

·         Define abbreviations at first use in the text (e.g., VEGF, PARP, EGF).

·         Figure 11 should be clear enough to the reader.

·         Table 2 is vague and should be revised for clarity.

·         Line 412-419: The conclusions would benefit from the authors giving more consideration to how the paragraphs are structured and the thesis of each paragraph. Please make it in a separate section.

Ref # 19-21 & 24 are very old- please update.

Author Response

Thank you very much for your valuable and insightful comments, which have greatly contributed to enhancing the quality of our manuscript. We have thoroughly and carefully revised the manuscript in accordance with your recommendations, with all changes highlighted in light gray. I hope these modifications meet the standards you have requested.

  1.  
  • Line 17: Clearly define “Xerostomia”.
  • Line 19: Define the study design here (e.g., This ? study investigated how…).
  • Line 23-29: Provide details about the statistical analysis used and significant p-value.

Reply. In the revised version, we added analysis of d-PAS positive goblet cell density, added “oral-gut axis” concept in the introduction part, and discussed the expression of VEGF, therefore, the abstract of this version was drastically changed. Due to the limitation of words counts for abstract, the first appearance of “Xerostomia” was in the introduction section on the line 56 and clearly defined it. The definition of the study and detailed statistical analysis methods were provided in abstract on the line 17-20, and 21-23, respectively.

  1. The introduction is very weak, fragmented, and not focused enough towards the research aim. The section should be qualified with more supporting evidence (e.g., Biores Open Access. 2017 Oct 1;6(1):123–132; J Oral Microbiol. 2022 Aug 11;14(1):2110194; Appl. Sci. 2020, 10, 6421). It is important to explore the mechanisms of microbial transmission, the impact of oral health on the gut microbiota alterations, and the potential role of dysbiosis in gut diseases both in vivo and in vitro/human studies. The main contribution or novelty needs a better explanation in the last paragraph.

Reply. Thank you very much for your insightful feedback. We have revised the Introduction to enhance its focus and provide a clearer point of view, particularly with respect to the "link of oral and intestinal microbiome". In doing so, we have carefully incorporated the perspectives and insights from the three manuscripts you recommended as reference 1,2 and 3, and reflecting the contents in the abstract (line 33), introduction (on the lines 41-45) and discussion part. This revision has allowed us to create more concise and precise Introduction that effectively guides readers into the context and objectives of our study.

  1. Line 437: Clear justification why these groups were selected? Define clearly sham and control groups.

Reply. The rationale for including these three groups—sialoadenectomy, sham, and control—was to control for potential confounding factors related to anesthesia, surgical stress, inflammatory responses, and the general surgical environment, all of which may independently influence intestinal conditions. By including sham group, we aimed to isolate the specific effects of sialoadenectomy from the broader physiological responses associated with surgery. This design allows us to more accurately attribute observed outcomes to the sialoadenectomy itself, thus enhancing the rigor and interpretability of our findings."

In response, we have clarified the definition of the three groups on lines 492-495.

  1. Line 466-469; Line 496-500; Line 546-563: The statistical analysis should be in a separate section and described in sufficient details.

Reply. All the statistical analysis were moved to “4.9 Statistical analysis” section.

  1. Line 532: Authors should clarify the rationale for using 16S rRNA Gene Amplicon Sequencing. This section should be described in more details.

Reply: The reason we chose 16S rRNA sequencing was as follows; it is a widely used for profiling microbial communities due to highly conserved 16S rRNA gene across bacteria, which lead to analyze at the genus or species level. We attempted to identify microbiota composition and relative abundance in samples which would be altered by sialoadenectomy. Therefore, we thoroughly rewrote “4.8 DNA extraction and 16S rRNA Gene amplicon” section with dividing 4 subsections on the lines 603-634.

4.8.1 DNA Extraction from Fecal Samples

4.8.2 16S rRNA Gene Amplification and Library Preparation

4.8.3 Sequencing on Illumina MiSeq Platform

4.8.4. Bioinformatics Analysis

  1. It is unclear what test used to analyze gene expression (e.g, VEGF).

Reply. We appreciate the opportunity to clarify the gene expression analysis conducted in this study. All gene expression measurements for EGF, PARP-1, and VEGF were assessed through immunohistochemistry (IHC) followed by semi-quantitative analysis using ImageJ software. Recognizing that our initial description may not have provided sufficient detail, we have revised Section 2.3 in the Results to divide into “d-PAS Staining” and “IHC” sections more distinctly as described on the lines 154-211.

Regarding VEGF expression, we observed no significant differences in semi-quantitative levels across the three groups over the experimental period. Consequently, we did not present a graph for VEGF and have clearly stated that “VEGF expression (Figure S2) remained strongly positive across all groups throughout the 12-month experimental period, with no significant differences observed in the semi-quantitative analysis (data not shown).” on lines 190-192. As you can see, other reviewer also mentioned about it. Furthermore, to enhance clarity, we have amended the subtitle of Section 4.7 to 'Immunohistochemistry and Semi-quantitative Analysis' to reflect the methodological approach used for gene expression analysis.

  1. Line 121: Please make “Length of jejunal and ileal villi of three groups” the title of table 1 (move to line 120).

Reply. All title for tables were transferred from the below to over each table.

  1. Define abbreviations at first use in the text (e.g., VEGF, PARP, EGF).

Reply. We defined all abbreviations for EGF, VEGF, and PARP at the first appearance on the lines 181-183 as follows: Epidermal Growth Factor (EGF), Vascular Endothelial Growth Factor (VEGF), and poly(ADP-ribose) polymerase-1(PARP-1).

  1. Figure 11 should be clear enough to the reader.

Reply. Upon reviewing, we found that the resolution of figures may have been inadvertently reduced during the file preparation process, particularly when copying and pasting or inserting JPEG files into the Word document. To address this issue, we added high-resolution versions of figure11 as supplementary material (Figure S4-S6) to ensure clarity and enhance the visual quality. We hope this resolves any concerns regarding figure quality.

  1. Table 2 is vague and should be revised for clarity.

Reply. Table 5 (previous Table 2) summarized commonly upregulated microbiome detected from all microbiome analysis- composition ratio, LEfSe and Heatmap analysis. To clarify the concept, the table was slightly updated as follows. I hope this modification will meet your standard.

Table 5: Summary of Commonly Upregulated Microbiomes Identified in Composition Ratio, LEfSe, and Heatmap Analysis by Group and Time Point

Group

Time

point

Upregulated microbiome

Sialoadenectomy

3M

o_Lactobacillales; f_Lactobacillaceae; g_Lactobacillus

Control

3M

f_Ruminococcaceae; g_Incertae_Sedis

3M

o_Peptostreptococcales-Tissierellales; f_Peptostreptococcaceae

3M

o_Oscillospirales; f_Ruminococcaceae; g_Romboutsia,

6M

f_Peptostreptococcaceae;g_Romboutsia

6M

o_Erysipelotrichales;f_Erysipelotrichaceae; g_Turicibacter

  1. Line 412-419: The conclusions would benefit from the authors giving more consideration to how the paragraphs are structured and the thesis of each paragraph. Please make it in a separate section.

Reply. This section was separated to “5. Conclusion” on the lines 665-672/

  1. Ref # 19-21 & 24 are very old- please update.

Reply. #19-21 were deleted during the modifying process for revised version, and #24 was replaced to #30.

Round 2

Reviewer 1 Report

Comments and Suggestions for Authors

The authors provided a very clear response to all the questions raised and made significant improvements to the manuscript.

Reviewer 3 Report

Comments and Suggestions for Authors

No further comments.